# Circularized fluorescent nanodiscs for probing protein–lipid interactions

Qian Ren[1,2], Shanwen Zhang[1,2] & Huan Bao [1✉]

Protein–lipid interactions are vital for numerous transmembrane signaling pathways. However, simple tools to characterize these interactions remain scarce and are much needed to advance our understanding of signal transduction across lipid bilayers. To tackle this challenge, we herein engineer nanodisc as a robust fluorescent sensor for reporting membrane biochemical reactions. We circularize nanodiscs via split GFP and thereby create an intensity-based fluorescent sensor (isenND) for detecting membrane binding and remodeling events. We show that isenND responds robustly and specifically to the action of a diverse array of membrane-interacting proteins and peptides, ranging from synaptotagmin and synuclein involved in neurotransmission to viral fusion peptides of HIV-1 and SARS-CoV-2. Together, isenND can serve as a versatile biochemical reagent useful for basic and translational research of membrane biology.

[1] Department of Molecular Medicine, UF Scripps Biomedical Research, 130 Scripps Way, Jupiter 33458 FL, USA. [2] These authors contributed equally: Qian Ren, Shanwen Zhang. ✉email: baoh@ufl.edu

Cells are compartmentalized by membranes and thus utilize transmembrane signaling to communicate with each other. To do so, membrane binding and remodeling reactions are involved in all aspects of cell-to-cell communication, ranging from synaptic transmission to cancer progression[1,2]. Therefore, extensive studies have been carried out to probe the molecular mechanism of membrane proteins that bind and reshape the structure of lipid bilayers[3–6]. Despite these efforts, simple and straightforward approaches to detect membrane binding and remodeling reactions are still lacking. Current approaches often rely on sophisticated equipment inaccessible to many labs[2]. To fill this gap, we herein seek to engineer a facile biosensor to monitor these membrane reactions.

GFP-based fluorescent sensors have been created for a multitude of applications[7–11]. In most sensor designs, the fluorophore of GFP is placed close to a sensing domain that could elicit a conformational change upon the binding of a specific substrate. The transduction of this conformational change to GFP would then result in a fluorescence readout. Inspired by these studies, we posit that a membrane binding and remodeling sensor is feasible if we could develop a strategy to bridge the conformational change of lipid bilayers and GFP.

On this front, nanodiscs could be a perfect system for grafting the sensing ability of GFP to the dynamics of membranes. Pioneered by the Sligar laboratory, nanodiscs encircle nanoscale lipid bilayers through membrane scaffold proteins (MSPs)[12,13]. Due to its excellent stability and homogeneity, nanodiscs have greatly facilitated structural and functional studies of membrane proteins in a lipid environment[14–16]. One intriguing observation from several of these efforts is that MSPs could adapt to the conformational states of membrane proteins enclosed in nanodiscs[17]. Thus, we hypothesize that MSP could serve as the bridge to connect the structural transition of lipid bilayers with the fluorescence emission of GFP. In this manuscript, we described our work in the development of an intensity-based fluorescent sensor for detecting membrane binding and remodeling events in nanodiscs (isenND), and demonstrated its use for the characterization of membrane proteins involved in a multitude of prokaryotic and eukaryotic transmembrane signaling pathways.

## Results

**Circularization of nanodiscs using split fluorescent proteins**. In light of recent fluorescent biosensor designs, we set out to fuse the two halves of split GFP (GFP$_{1-10}$/GFP$_{11}$) to the N- and C-termini of MSP1D1 that would form a 10 nm nanodisc, generating spGFP$_{1-10/11}$-MSP1D1 (Fig. 1a)[18]. The idea is that movement of MSP upon the remodeling of lipid bilayers would alter the local environment of the chromophore of GFP and thus result in its fluorescence change[19]. In addition, complementation of split GFP would serve as a simple method to circularize nanodiscs for enhanced yield, stability and monodispersity.

The fused protein spGFP$_{1-10/11}$-MSP1D1 expressed well in bacterial cells and recovered the fluorescence of a typical GFP (Fig. 1b). Using size-exclusion chromatography (SEC), we found that most purified proteins were monomeric formed by intramolecular complementation, with ~30% are dimeric formed by intermolecular complementation (Fig. 1c). Upon reconstitution with lipids using the monomer, nanodiscs were readily formed and purified by SEC (Fig. 1c). Consistent with previous studies[12,13], dynamic light scattering (DLS) and negative-stain electron microscopy (EM) experiments further demonstrated the expected ~12 nm diameter of the spGFP$_{1-10/11}$-MSP1D1 nanodiscs (Fig. 1d, e and Supplementary Fig. 1). Thus, split GFP complementation is amenable to MSP circularization, which is

critical to construct much larger 30 and 50 nm nanodiscs for biochemical reconstitution of complex membrane systems[20–22]. We, therefore, fused the split GFP$_{1-10/11}$ onto NW30 and NW50 that could encase these large nanodiscs upon circularization (Fig. 1b). Using SEC and EM analysis, we indeed found that spGFP$_{1-10/11}$-NW30 and spGFP$_{1-10/11}$-NW50 could form 29 and 48 nm large circularized nanodiscs (Fig. 1c–e), respectively. These values were also corroborated by DLS measurements (Supplementary Fig. 1). Similar to spGFP$_{1-10/11}$-MSP1D1 nanodiscs, intermolecular complementation of spGFP$_{1-10/11}$-NW30 and spGFP$_{1-10/11}$-NW50 was also observed, as evidenced by the broader SEC profile as compared to our previous nanodiscs[22].

To expand the palette of isenND, we sought to circularize nanodiscs using split red fluorescent proteins (RFPs) such as mcherry, generating spmcherry$_{1-10/11}$-MSP1D1 (Supplementary Fig. 2a, b)[23]. To our surprise, none of the split mcherry in our trials complemented when fused to the N- and C-termini of MSP1D1 (Supplementary Fig. 2c–e). Furthermore, fusion of several full-length RFPs with MSP was also not successful (Supplementary Fig. 2c–i). These problems could arise due to the much lower stability of RFP as compared to GFP[23]. To tackle this challenge, we employed the SpyCatcher-SpyTag-mediated protein circularization approach to promote the formation of spmcherry$_{1-10/11}$-MSP1D1 (Supplementary Fig. 2b; SPY-spmcherry$_{1-10/11}$-MSP1D1)[24–26], as our recent work found that covalently cyclized MSPs exhibited greatly improved solubility and stability[22]. Using this approach, we successfully generated the RFP-circularized MSP1D1 (Fig. 1b and Supplementary Fig. 2c, j and k; SPY-spmcherry1$_{1-10/11}$-MSP1D1), which could form homogenous 12 nm nanodiscs as shown by SEC and DLS characterizations (Supplementary Fig. 3). Moreover, the addition of the SpyCatcher and SpyTag enforced the intramolecular complementation of mcherry, as the dimer formation of the scaffold protein from intermolecular complementation was less than 10% (Supplementary Fig. 3a). We thus also fused the SpyCatcher and SpyTag onto spGFP$_{1-10/11}$-MSP1D1 and indeed obtained nanodiscs with increased monodispersity (Supplementary Fig. 3b, c). The efficiency of SPY- spmcherry1$_{1-10/11}$-MSP1D1 to form nanodiscs is lower than SPY-spGFP$_{1-10/11}$-MSP1D1, as more than 35% larger particles were found by SEC analyses (Supplementary Fig. 3a and b). Nevertheless, complementation of split fluorescent proteins can circularize nanodiscs, paving the way for the development of the isenND toolkit for reporting membrane binding and remodeling reactions.

**Using isenND to detect membrane binding**. Every membrane remodeling reaction starts from a protein binding to lipids. With the readily available GFP on nanodiscs, we should be able to characterize protein-membrane interactions via FRET (Fig. 2a). We tested this idea using the cytosolic domain of synaptotagmin-1 (syt1)[27,28], the Ca$^{2+}$ sensor for synaptic transmission[29,30]. We fused syt1 with mcherry and examined its binding to spGFP$_{1-10/11}$-MSP1D1 nanodiscs bearing PS lipids. The fluorescence spectrum revealed a significant amount of FRET that was further enhanced by the addition of Ca$^{2+}$ (Fig. 2b). In control experiments, we did not observe FRET using nanodiscs harboring only PC lipids (Fig. 2c), which is consistent with the preferred binding of syt1 to negatively charged lipids[31,32].

In addition, spGFP$_{1-10/11}$-MSP1D1 nanodiscs can allow us to characterize syt1-lipid interaction via gel-electrophoresis (Supplementary Fig. 4). Here, spGFP$_{1-10/11}$-MSP1D1 nanodiscs were incubated with increasing concentrations of syt1. Binding of syt1 caused a mobility shift of nanodiscs on native PAGE (Supplementary Fig. 4a), which could be rapidly quantified by in-gel

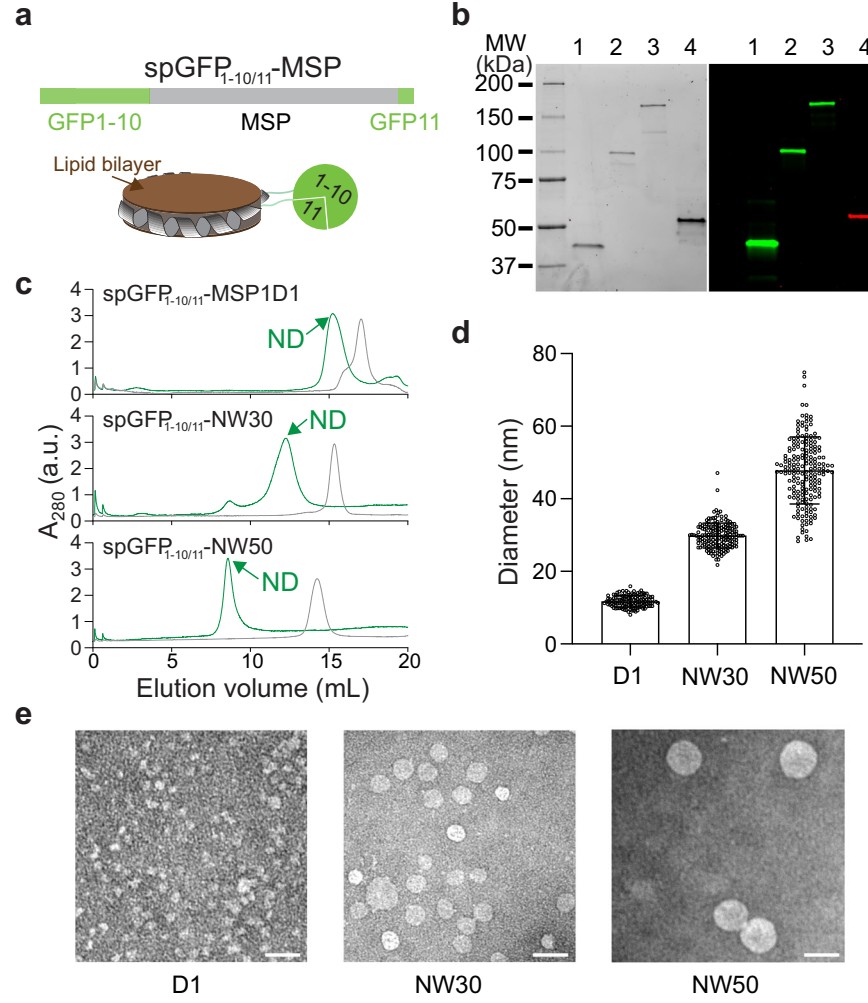

**Fig. 1 Circularization of nanodiscs by GFP complementation. a** Illustration of isenND. Split GFP enables the circularization of the spGFP$_{1-10/11}$-MSP1D1 protein (top) and nanodisc (bottom). **b** SDS-PAGE (left, Coomassie-blue stained) and in-gel fluorescent imaging analysis (right) of circularized MSPs. **1**, spGFP$_{1-10/11}$-MSP1D1; **2**, spGFP$_{1-10/11}$-NW30; **3**, spGFP$_{1-10/11}$-NW50; **4**, SPY_spmcherry1$_{1-10/11}$-MSP1D1. **c** Representative SEC profiles of the indicated MSPs (gray) and nanodiscs (ND, green). For nanodisc reconstitution, protein:lipid ratios were: 1:60 for spGFP$_{1-10/11}$-MSP1D1, 1:200 for spGFP$_{1-10/11}$-NW30, 1:400 for spGFP$_{1-10/11}$-NW50. **d** Diameters of split GFP circularized nanodiscs determined by negative-stain EM. **D1**, spGFP$_{1-10/11}$-MSP1D1; **NW30**, spGFP$_{1-10/11}$-NW30; **NW50**, spGFP$_{1-10/11}$-NW50. Data are shown as mean ± s.d. from multiple measurements using $n \geq 3$ independent sample preparations. **e** Representative EM micrographs of split GFP circularized nanodiscs. Scale bar, 50 nm.

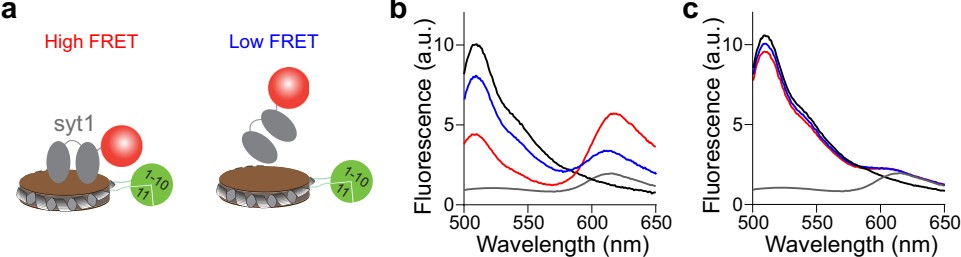

**Fig. 2 Using isenND to characterize protein-lipid interactions. a** Diagram for the FRET between syt1-mcherry and isenND. **b**, **c** Representative fluorescence spectrum of syt1-mcherry (gray) binding to isenND (black). FRET was observed upon incubation of syt1-mcherry with isenND containing PS lipids in the presence (red) and absence (blue) of 0.5 mM Ca$^{2+}$ (**b**). In contrast, FRET was not observed between syt1-mcherry and isenND containing only PC lipids (**c**). Data are shown as mean ± s.d., $n = 3$ independent experiments.

fluorescence imaging at the GFP channel (Supplementary Fig. 4b). Consistent with the FRET-based assay, syt1 binds with a much higher affinity to PS lipids than PC. Thus, isenND is able to capture the association of peripheral membrane proteins with lipids.

**Engineering isenND to detect membrane remodeling**. Next, we sought to explore if isenND can sense the remodeling of lipid bilayers. As a proof-of-principle study, we again used syt1 as it is known to bend the plasma membrane for vesicle exocytosis[33–36]. GFP$_{1-10/11}$ circularized nanodiscs were not able to sense the

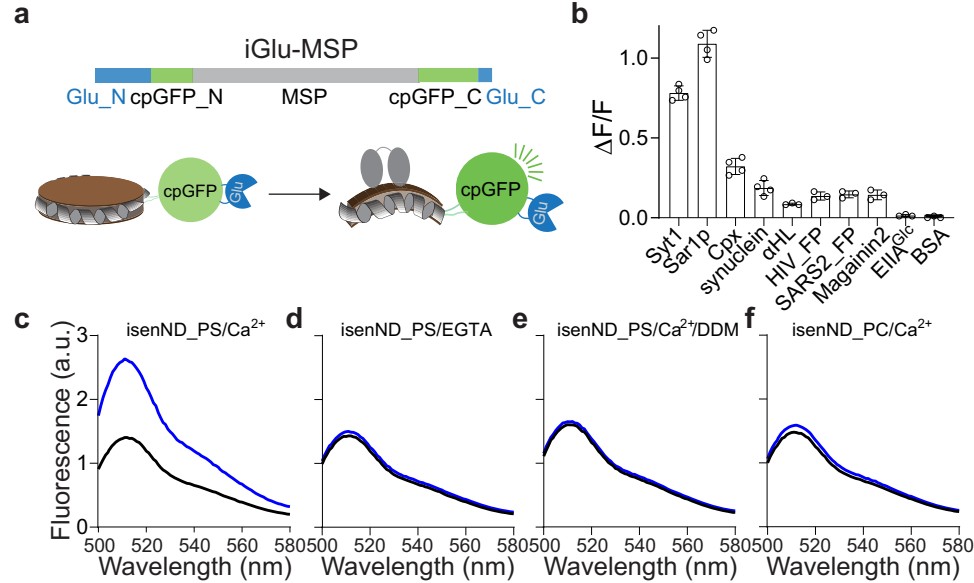

**Fig. 3 Engineering isenND to sense the membrane remodeling reaction. a** Illustration of iGlu-MSP. Top, construct design. Bottom, binding of syt1 bends the membrane in isenND and increases its fluorescence. **b** Sensitivity of iGlu-MSP nanodiscs for different membrane remodeling proteins. iGlu-MSP nanodiscs (0.1 μM) were incubated with the indicated protein (1 μM) at room temperature for 10 min, followed by fluorescence spectroscopy analyses. EIIA$^{Glc}$ and BSA were used as controls. Nanodiscs were prepared using 50% PC, 40% PS and 10% PE (isenND_PS) unless otherwise specified. The cytosolic domain of synaptotagmin-1 (syt1), 37 kDa; sar1p, 21 kDa; complexin-2 (cpx), 15 kDa; α-synuclein (syn), 14 kDa; α-hemolysin (αHL), 36 kDa; HIV-1 fusion peptide (HIV_FP), 2.9 kDa; SARS-CoV-2 fusion peptide (SARS_FP), 2.5 kDa; magainin2 (Mag2), 2.5 kDa; EIIA$^{Glc}$, 22 kDa; BSA, 67 kDa. **c–f** Representative fluorescence spectrum of iGlu-MSP nanodiscs in the absence (black) and presence (blue) of syt1 supplemented with 0.5 mM Ca$^{2+}$ (**c**), 1 mM EGTA (**d**), or 0.5 mM Ca$^{2+}$ followed by 0.2% DDM (**e**). In the control experiment (**f**), iGlu-MSP nanodiscs contaning only PC lipids (isenND_PC) were used. Data are shown as mean ± s.d., $n \geq 3$ independent experiments.

action of syt1 (Supplementary Fig. 5a), suggesting the force generated from the membrane bending reaction by syt1 is not sufficient to reconfigure the fluorophore in the complemented GFP$_{1-10/11}$.

We then tested split circularly permuted GFP (spcpGFP)[8], as its fluorophore would be closer to MSP and has been successfully used to develop a multitude of biosensors[7–11]. Even though circularized nanodiscs could be formed using spcpGFP-MSP1D1, syt1-mediated membrane remodeling reactions were still not detectable (Supplementary Fig. 5b). We suspect that this result was due to the inherent flexibility of nanodiscs. To bypass this limitation, we again used the SpyCatcher-SpyTag to enhance the rigidity of nanodiscs (Supplementary Fig. 5c). The resulting SPY-spcpGFP-MSP1D1 nanodiscs responded to the syt1-mediated membrane remodeling reaction with a ΔF/F of 0.1. This low signal-to-noise ratio might come from the perhaps too rigid framework coined by the SpyCatcher-SpyTag. Therefore, we replaced it with the two halves of the glutamate binding protein (Fig. 3a), which was used to build a robust glutamate sensor iGluSnFR[7,8]. After screening a few different linkers between cpGFP and MSP1D1, the resulting construct iGlu-MSP1D1 indeed formed nanodiscs that exhibited an excellent sensitivity to the membrane remodeling reaction by syt1 with a ΔF/F of 0.8 (Fig. 3b, c and Supplementary Fig. 5d), which was dependent on Ca$^{2+}$ (Fig. 3d) and could be diminished by the addition of 0.2% DDM (Fig. 3e). Further, C2A and C2A-C2B$_{CLM}$, mutants of syt1 that were much less efficient in bending membranes[34], resulted in significantly lower fluorescence responses (Supplementary Fig. 5e). In contrast, C2B, which retained the ability to bend membranes, caused a high fluorescence response as the wild-type protein. Interestingly, previous studies showed that C2A$_{CLM}$-C2B is less active in membrane bending than the wild-type protein using negative stain EM[34], while exhibiting similar responses in our assay. We suspect that this discrepancy is most likely due to

the sensitivity of the two assays, although the exact reason is unclear and will require structural elucidation of membrane bending by syt1. Nevertheless, consistent with the preferred binding of syt1 to negatively charged lipids, iGlu-MSP1D1 nanodiscs made with only PC lipids showed a ΔF/F less than 0.1 in the presence of syt1 and Ca$^{2+}$ (Fig. 3f). We also found that the sensitivity of these nanodiscs could tolerate low amounts of DDM, perhaps because of the enhanced rigidity of the circularized scaffold protein (Supplementary Fig. 5f).

We further tested the performance of iGlu-MSP1D1 nanodiscs for a panel of proteins that could remodel membrane, ranging from synaptic proteins to viral fusion peptides (Fig. 3b). Gratifyingly, the data revealed that iGlu-MSP1D1 nanodiscs were able to detect membrane remodeling actions of all of these proteins, including vesicle budding protein sar1p[37], α-synuclein aggregates[38], pore-forming toxin α-hemolysin[39], antibacterial peptide magainin2[40], viral fusion peptides from HIV-1[41] and SARS-CoV-2[42]. In control experiments, the sar1p mutant (T39N) that was dominant negative in mediating vesicle budding[43], exhibited a ~6-fold decrease in the sensor response (Supplementary Fig. 5e). In general, the responses of the sensor to proteins that disrupt membranes (e.g., toxins) are much smaller than those bend membranes (e.g., syt1 and sar1p). Interestingly, the sensor indicated that the synaptic protein complexin[44–48] (Cpx) reshaped membranes (Fig. 3b). This remodeling activity lies in its C-terminal amphipathic helix, which is also predicted to have membrane sculpting activities as antibacterial peptides using a computational algorithm (Supplementary Fig. 6a)[49]. Indeed, we confirmed these predictions of complexin using biochemical reconstitutions (Supplementary Fig. 6b–e). Both full-length complexin and a peptide corresponding to the C-terminal amphipathic helix (C21) markedly caused membrane leakage and deformation (Supplementary Fig. 6b–d), whereas the complexin mutant lacking C21 showed little effect (Supplementary Fig. 6b–d,

CpxΔC21). This membrane remodeling activity is also essential for complexin to promote vesicle fusion (Supplementary Fig. 6e), perhaps by lowering the energetic requirement through membrane remodeling to merge two opposing lipid bilayers[50]. These interesting findings warrant further dissection of the function of complexin in future studies. In control experiments, we did not detect much membrane remodeling activity of a bacterial peripheral protein EIIA$^{Glc}$ (Fig. 3b), which is able to bind lipids in nanodiscs[22,32,51]. Together, we concluded that the isenND toolkit could confer versatile interrogations of membrane binding and remodeling events involved in a multitude of transmembrane signaling pathways.

## Discussion

In this manuscript, we developed a simple yet robust fluorescent reporter (isenND) by leveraging the sensing power of GFP with the elegant framework of nanodiscs for probing membrane binding and remodeling reactions. We showed that isenND could rapidly profile the action of a variety of membrane proteins on lipid bilayers within a few minutes. In contrast, traditional approaches to assay membrane binding and remodeling reactions often involve giant unilamellar vesicles (GUV) and microscopy[2]. Generation of GUV is not trivial and high-throughput screens using GUV entail sophisticated equipment and expertise. Importantly, GUVs are not stable and thus need to be freshly prepared before each experiment. As such, this type of experiment is challenging to perform, labor-intensive and time-consuming. Here, our isenND sensor could be easily produced in large quantities and stably stored in a -80 °C freezer until use. In addition, the sensor is able to report the membrane bending reaction by several proteins that are difficult for GUV-based assays[33–36]. Most importantly, the use of isenND only requires basic equipment to monitor fluorescence emission and is compatible with most high-throughput plate readers. Thus, we believe that our tools could greatly simplify the study of protein-lipid interactions, thereby enabling facile characterizations of membrane binding and remodeling proteins.

Moreover, we have uncovered another new approach to circularize nanodiscs using split GFP. Previous studies employed the sortase-mediated protein conjugation method to circularize MSPs in vitro[20]. In this approach, the MSPs for the construction of large circularized nanodiscs were mostly insoluble and degraded when overexpressed in cells and thus usually have much lower yields as compared to the scaffold protein for small nanodiscs. Additional purification and circularization reactions in vitro are laborious and would further decrease the yield. These challenges impeded the widespread application of circularized large nanodiscs, even though they are essential for the structural and functional studies of several complex membrane biochemical reactions. In our work, MSPs were circularized via split GFP complementation immediately after translation in cells, which could protect the hydrophobic face of MSP and thus result in much-improved yields and simplified procedures for the construction of circularized nanodiscs.

In summary, we herein eased the production of circularized nanodiscs and further functionalized them as fluorescent reporters. The resulting toolkit, isenND, could swiftly determine if the protein of interest would bind and sculpt the lipid bilayer. Since many membrane binding and remodeling reactions are drug targets, these advantages bode well for the further development of isenND into biochemical reagents useful in both basic and translational research of membrane biology.

## Methods

**Chemicals and reagents**. 1,2-dioleoyl-sn-glycero-3-phosphocholine (PC), 1,2-diphytanoyl-sn-glycero-3-phosphocholine (DPhPc), 1,2-dioleoyl-sn-glycero-3-phospho-l-serine (PS) and 1-palmitoyl-2-oleoyl-sn-glycero-3-phosphoethanolamine (PE) were obtained from Avanti Polar Lipids. His60 Ni Superflow Resins were purchased from Takara Bio USA and Superose 6 Increase 10/300 GL was from GE Healthcare. All other chemicals were acquired from Sigma.

**Plasmids**. pET28a-MSP1D1 was a gift from Dr. Steven Sligar[12]. pET28a-NW30 and pET28a-NW50 were gifts from Dr. Gerhard Wagner[20]. pET_sfCherry(1-10)_32aalinker_sfCherry(11) was a gift from Dr. Bo Huang[23] (Addgene plasmid # 83030). iGluSnFR was a gift from Dr. Loren Looger[7,8]. pET28a-mcherry was a gift from Dr. Scott Gradia (Addgene plasmid # 29769). CH-GECO2.1 (Addgene plasmid # 52099), pBAD-RDSmCherry1 (Addgene plasmid # 89987), pBAD-LSSmCherry1 (Addgene plasmid # 89986) were gifts from Dr. Robert Campbell[52,53]. pGEX2T-sar1p was a gift from Dr. Anjon Audhya[54]. All other constructs in this work were made using the In-Fusion® HD Cloning Kit (Takara Bio USA). The protein sequences of the engineered MSPs are described in Supplementary Table 1.

**Proteins and peptides**. syt1, cpx2, sar1p, synuclein and SNAREs were expressed in BL21 STAR™ (DE3) and purified using GSTrap and Ni$^{2+}$-NTA columns[54–60]. α-hemolysin was purchased from Sigma. HIV-1 and SARS-2 fusion peptides were synthesized from Genscript: AVGIGALFLGFLGAAGSTMGAASGGGKKKKK and KQYGDCLGDIAARDLICAQKFNG. For the production of MSPs characterized in this study, plasmids were transformed into BL21 STAR™ (DE3) cells that were grown in LB supplemented with Km (50 mg/mL) to OD$_{600}$ ~0.5. Protein expression was induced with 0.1 mM IPTG at 16 °C overnight. Bacteria were harvested by centrifugation at 3700 rpm for 20 mins, resuspended in Buffer A (50 mM Tris-HCl (pH 8),100 mM NaCl, 10% glycerol, 2 mM β-mercaptoethanol), and lysed using a Branson cell disrupter. Cell lysates were clarified by centrifugation at 10,000 rpm for 1 h. The supernatants were incubated with 1 mL His60 Ni Superflow Resins for 20 min with gentle shaking at room temperature. Samples were loaded onto an Enco gravity column (Bio-Rad # 9704652), followed by two times wash using buffer B (50 mM Tris-HCl (pH 8), 30 mM Imidazole, 500 mM NaCl, 10% glycerol, 2 mM β-mercaptoethanol). Proteins were eluted in buffer C (50 mM Tris-HCl (pH 8), 500 mM Imidazole, 500 mM NaCl, 10% glycerol, 2 mM β-mercaptoethanol), desalted in buffer A using PD MiDiTrap G-10 (GE Healthcare), and stored at −80 °C.

**Nanodiscs**. Lipids were dried under a stream of nitrogen and resuspended in Buffer D (20 mM Tris-HCl (pH 8), 50 mM NaCl and 1 mM DTT). For nanodisc reconstitution, MSPs were incubated with lipids (50% PC, 40% PS and 10% PE) at the indicated ratios in buffer A containing 0.05% DDM. Detergents were slowly removed by gentle shaking with BioBeads (4 °C, overnight). Samples were purified by gel filtration using Superose 6 10/300 (GE Healthcare) in buffer A and stored at −80 °C.

**Pull-down assays**. To screen the expression level of MSPs, cells were grown in 20 mL culture to OD600 ~0.5 and induced with 0.1 mM IPTG at 16 °C overnight. Bacteria were harvested by centrifugation at 6000 rpm for 10 mins, resuspended in Buffer A and lysed by freeze-thaw plus the addition of 1% Triton at 4 °C overnight. Cell lysates were clarified by centrifugation at 10,000 rpm for 20 mins. The supernatants were incubated with 100 μL His60 Ni Superflow Resins for 20 min with gentle shaking at room temperature. Samples were loaded onto empty micro Bio-Spin chromatography columns (Bio-Rad #7326204), followed by two times wash using buffer B. Proteins were eluted in buffer C and subjected to analysis by SDS-PAGE.

**Fluorescence spectroscopy**. Proteins or peptides (1 μM) were incubated with nanodiscs (100 nM) in assay buffer (20 mM Tris-HCl (pH 8), 50 mM NaCl and 0.5 mM DTT). The fluorescence spectrum of GFP-labeled isenND was collected on a Synergy H1M plate reader (BioTeK) with excitation at 460 nm and emission from 500–650 nm. For RFP-labeled isenND, samples were excited at 560 nm and emission was collected from 600-700 nm.

**Liposome leakage assay**. Lipids (70% PC and 30% PS) in chloroform were dried under a gentle stream of argon and further in a vacuum for 3 h. The dried lipid films were resuspended in buffer D plus 50 mM glutamate at room temperature, followed by extrusion through a 100 nm polycarbonate filter (Whatman) 30 times using the Mini-Extruder device (Avanti Polar Lipids). Liposomes encapsulated with glutamate were purified using PD-10 desalting columns (GE Healthcare) equilibrated in buffer D and were used immediately for the leakage assay by addition of complexin at the indicated concentrations along with the glutamate sensor iGluSnFR[7] (1 μM). Glutamate leakage from liposomes was monitored using a Synergy H1M plate reader with excitation at 460 nm and emission at 520 nm. After 30 mins, 0.25% DDM was added to each reaction and data were collected for another 10 mins. The percentage of glutamate leakage was normalized against the maximal fluorescence signal after the addition of DDM. Data were obtained from three independent trials.

**Spheroplast lysis assay**. *E.coli* DH5α cells were grown to $OD_{600}$ ~0.5 and resuspended in buffer E (20 mM Tris-HCl (pH 8), 100 mM NaCl, 18% sucrose, 2 mM EDTA). By incubation with lysozyme (50 µg/mL) on ice for 1 h, spheroplasts were generated from these cells and then sedimented by centrifugation at 2000 rpm for 20 min at 4 °C after two washes. Spheroplasts were used immediately for the lysis assay by 10-fold dilution in buffer E with or without the wild type and variants of complexin at the indicated concentrations. The lysis of spheroplasts was monitored by measuring the absorbance at 500 nm via a Synergy H1M plate reader. After 30 min, the absorbance values were used to plot the data as a function of protein concentrations. Data were obtained from three independent trials.

**Fusion assay**. SNARE proteins were reconstituted into proteoliposomes using a detergent-assisted approach[58,59]. Briefly, lipids and SNARE proteins were incubated in buffer A supplemented with 0.1% DDM. Detergents were slowly removed by addition of BioBeads (1/3 volume) and gentle shaking (4 °C, overnight). Liposomes harboring SNAREs were extruded through an Avanti extruder with 200 nm filter and further purified using PD MiDiTrap G-25 (GE Healthcare) in buffer A. v-SNARE vesicles were made with lipids composed of 27% PE, 25% PS, 45% PC, 1.5% NBD-PE and 1.5% rhodamine-PE, whereas t-SNARE vesicles were made with 30% PE, 25% PS, 45% PC. The lipid mixing assays were carried out by incubation of v- and t-SNARE vesicles in the presence of syt1 (1 µM) and $Ca^{2+}$ (0.5 mM) in buffer D plus the macromolecular crowding agent[61,62], Ficoll70 (50 mg/mL). The NBD signal was monitored using a Synergy H1M plate reader with excitation at 460 nm and emission at 530 nm. After 60 min, the efficiencies of membrane fusion were determined by normalization of data to the maximal fluorescence signal after the addition of 0.5% DDM to each sample.

**Planar lipid bilayer electrophysiology**. Experiments were performed on an Orbit Mini apparatus (Nanion) using MECA chips (100 µm) in 20 mM Tris-HCl, pH 8, 100 mM NaCl. The lipid bilayer was painted with DPhPc at 5 mg/mL in n-octane. Purified complexin and mutants were added at 10 µg/mL. Once single pore-formation events were detected, a customized perfusion system (Eastern Scientific LLC) was used to slowly remove excess proteins. All electrical recordings were collected at 1.25 kHz, filtered at 625 Hz and analyzed in Clampfit 10.

**Statistics and reproducibility**. GraphPad Prism 9 is used to analyze data generated in this study. Data are presented as mean ± s.d. Independent experiments were defined as replicates performed with separately prepared batches of protein and nanodiscs.

**Other methods**. SDS-PAGE was performed using 4–15% TGX protein gels (Bio-Rad) and imaged on a GelDoc Go system (Bio-Rad). Size-exclusion chromatography (SEC) was carried out using a Superose 6 Increase 10/300 GL column equilibrated in 50 mM Tris-HCl, pH 8, 100 mM NaCl, 5% Glycerol on an AKTA pure 25 L (GE Healthcare). Dynamic light scattering was performed on a DynaPro NanoStar instrument (Wyatt Technology)[56,57]. Electron microscopy of NDs stained with uranyl formate was carried out using Formvar/carbon-coated copper grids (01754-F, Ted Pella, Inc.) on a ThermoFisher Science Tecnai G2 TEM (100 kV) equipped with a Veleta CCD camera (Olympus)[22,63,64].

**Reporting summary**. Further information on research design is available in the Nature Research Reporting Summary linked to this article.

## Data availability
Gel images and fluorescence spectroscopy data are provided in the Source Data file and all other data are included in the paper. All the plasmids will be available at Addgene. Source data are provided with this paper in Supplementary data 1.

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

## Acknowledgements
We thank Dr. Michael Farzan for access to the DLS instrument. We thank Drs. Naomi Kamasawa and Debby Guerrero-Given from the imaging center at the Max Planck Florida Institute for Neuroscience for assist in EM. We would also like to thank Dr. Franck Duong for the plasmid of pBad33-EIIA$^{Glc}$, Dr. Anjon Audhya for the plasmid pGEX2T-sar1p, and Dr. Edwin Chapman for the plasmids of pGEX4T-syt1 and pGEX4T-cpx2. This work is supported by the NIH Director's New Innovator Award (DP2GM140920 to H.B.). Work shown in Supplementary Fig. 6b and c was carried out at UW-Madison and supported by Human Frontier Science Program postdoctoral fellowship no. LT000712 (Huan Bao), Pew Charitable Trust grant no. 864K625 (Edwin R. Chapman and Dorit Hanein) and National Institute of Health grant nos. MH061876 and N097362 (Edwin R. Chapman).

## Author contributions

Q.R. and S.Z. performed the experiments and analyzed the data. H.B. conceived the project and wrote the manuscript.

## Competing interests
The authors declare no competing interests.
