## [Peer Review File · Communications Biology]

Reviewers' comments:

Reviewer #1 (Remarks to the Author):

The author uses a newly developed nanodisc combined with a fluorescent sensor to detect membrane biochemical changes. It is interesting and getting a promising result. However, this resulting lack of specific control experiments must be repeated in the exact experimental setting. My suggestion is author revise the manuscript and include the following experiment result to clarify the major conclusion:

1. cryoEM or TEM to further validate nanodisc size determined by DLS and SEC.
2. Control experiments demonstrate that the membrane does undergo changes/bind and remodeling under authors' experimental settings. The author designed a new nanodisc system connecting GFP or other components. This may already change the membrane conditions or structures of the nanodisc. Does the membrane structure change as the author expected?
3. What happens when the different concentrations of detergent are mixed in the buffer? A typical experiment of nanodisc replacement may still contain detergent in low concentration. Dose it will affect membrane formation or bind associated with the membrane?

Reviewer #2 (Remarks to the Author):

Over the last decade, nanodiscs (NDs) have emerged as a powerful approach to reconstitute membrane proteins in a lipid environment. NDs are nanosized objects compatible with a large panel of applications. They have inspired many research groups to improve and optimize membrane scaffold proteins (MSPs) in order to make this membrane mimicking system even more versatile.

In their study, Qian Ren and colleagues followed this path by designing circularized fluorescent NDs with the aim of developing new biosensors to monitor membrane binding and remodeling events. Although covalently-circularized NDs were already introduced a few years ago to stabilize large NDs, this study reports the complementation of the two halves of split GFP fused to the N- and C-termini of MSPs. Three MSPs of different lengths were used to generate circularized fluorescent NDs of different diameters. In order to link the GFP fluorescence emission signal to the membrane deformation mediated by protein binding to NDs, the authors optimized the GFP-MSP constructs. Then, as a proof of concept, they tested a panel of proteins that are capable of membrane remodeling.

In addition to the development of these new NDs, the improvement of circularized NDs, which are stable and homogenous contributes to an improvement of this tool for the study of membrane proteins. Overall, the manuscript is clear and well written but many experimental details and clarifications in the figure legends deserved to be added. My main comments are listed below:

- It is reported that all attempts to generate RFP-MSPs (via split RFPs or full-length RFPs) failed, unless SPY-spmcherry was used. The authors interpreted this result as a problem with RFP stability. However, it is unclear if the problem arises from a lack of RFP-MSP expression. It would have been interesting to indicate the levels of expression of the different constructs. In the case of satisfactory expression levels of RFP-MSP, their ability to form NDs in the presence of lipids would be worth mentioning.

- In Fig. S1L, only the SEC profile of SPY-spSFM-MSP (grey curve) has been commented to mention the low proportion of dimers (less than 10%). However, no comment is made on the profile of RFP-NDs (red curve) although it seems to show a higher proportion of dimers. Further comments on the results deserve to be given.

- Based on the experiment performed in Fig. 2, the authors seem to suggest that the increase in FRET signal between PS containing GFP-ND and syt1-mcherry in the presence of calcium ions (Fig. 2B) results from a specificity of syt1 to PS since the FRET signal obtained in the presence of PC containing GFP-ND is much weaker. However, knowing that PS is negatively charged and PC is

zwitterionic, calcium ions act as bridges and the experiment performed in Fig. 2C with PC cannot be interpreted as a control experiment to reveal syt1 specificity to PS. An increased in FRET signal would have been observed with any negatively charged surface. One note that the concentration of Ca²⁺ used is not reported in the Figure legend or in the Method section. More experimental details would be appreciated.

- It is mentioned that different linkers between cpGFP and MSP have been screened in order to optimize NDs with the expected sensitivity to membrane protein binding. The given explanations appear rather speculative and although we don't really know why it ends up working, it seems to remain a black box. Here again, based on the level of detail provided in the Method section it appears impossible to reproduce the cloning work. More information on the sequences used for the iGlu-MSP construction is required to make this study of interest for other researchers.

Other minor comments:

Globally, the figure legends are not precise enough.

- Figure 1B: mention in the legend if the gel is Coomassie-blue stained.

- Figure S2A: precise in the legend that the gel was illuminated at the GFP channel for in-gel fluorescent signal detection. Indicate if the experiment has been done in the presence of calcium. As syt1-ND shows smears on the gel, indicate with a box the area that has been used for signal quantification.

- Figure 3B: Precise in the legend the concentrations of GFP-NDs and that of membrane remodeling proteins. The addition of the full name of these latter and their molecular weight could be informative.

- Figure 3C: for the blue and black curves present on the fluorescent spectra, specify the conditions to which they are associated.

Reviewer #3 (Remarks to the Author):

Ren et al., describe a fluorescence-based assay for detecting membrane protein binding events on circularised nanodiscs. This methodology which relies on split-GFP technology could also be used to circularise (and detect formed) nanodiscs could constitute a standard tool in membrane protein biophysics and structural biology. The method could be used for sample condition, or protein reconstitution optimisation and to monitor biological (related to membrane binding) processes. The authors test their isenND method on various pro- and eukaryotic proteins, and demonstrate its ease of use and applicability. They also highlight the potential of their method as being fast and reliable, relying on readily accessible and affordable equipment.

The manuscript is well written and I see no major flaws with neither the reported data reproducibility or its analysis. Circularised nanodiscs are becoming increasingly important in membrane protein biochemistry, biophysics and structural biology. The pursue for highly stable nanodiscs to accommodate large protein complexes for analysis by CryoEM, EPR (PELDOR or DEER) or smFRET, makes this method an important tool for modern structural biology, where there is now an increasing need for studying membrane proteins in their native lipid environment and thus acquiring data within stable, monodisperse, circularised nanodiscs of any size. In particular, for larger nanodiscs described here (NW30/50), which are more challenging to form, stabilise and control, due to their highly dynamic nature, a tool such as this, could allow for the detection of specific protein binding events and interactions with membranes, using readily accessible equipment.

Points to address:

- 1) On figure 1C (bottom). If NW50 elute at "void" volume of a Superose 6 column, how could the NW50 monodispersity be judged? Perhaps another column with lower resolution (for larger particles though) may be able to discriminate among NW50 sizes/states, e.g. Sephacryl S-500? The authors present supporting DLS data (1D) for NW50 size distribution, however it would be good to support DLS conclusions with higher resolution SEC, given that "large" NW50 nanodisc formation is more of a "bigger" issue (significantly broader distribution in DLS), compared to forming smaller nanodiscs. Is it possible that for NW50 two or more species co-exist? If yes, this could have implications for high resolution structural studies, such as cryoEM.
- 2) On figure 2. Could the authors clarify what the "independent" experiments mean? Independent nanodisc batches for each, protein preparations, etc.
- 3) Again in Figure 3 as in figure 2 (previous comment). Please specify what "independent" experiments account for

Reviewer #1 (Remarks to the Author):

The author uses a newly developed nanodisc combined with a fluorescent sensor to detect membrane biochemical changes. It is interesting and getting a promising result. However, this resulting lack of specific control experiments must be repeated in the exact experimental setting.

Response: We thank this reviewer for the positive feedback and have performed the requested control experiments in the revised manuscript. We hope the reviewer finds these revisions satisfactory.

My suggestion is author revise the manuscript and include the following experiment result to clarify the major conclusion:

1. cryoEM or TEM to further validate nanodisc size determined by DLS and SEC.

Response: We have now analyzed these nanodiscs using negative stain EM (new Fig. 1D and E) and found that the size distributions of nanodiscs from these EM analyses were similar to DLS measurements. In the revised manuscript, we have moved the previous DLS data into the new Fig. S1 and made the following changes:

“Consistent with previous studies^{12, 13}, dynamic light scattering (DLS) and negative-stain electron microscopy (EM) experiments further demonstrated the expected ~12 nm diameter of the spGFP_{1-10/11}-MSP1D1 nanodiscs (Fig. 1D, E and S1).”

“These values were also corroborated by DLS measurements (Fig. S1).”

2. Control experiments demonstrate that the membrane does undergo changes/bind and remodeling under authors' experimental settings. The author designed a new nanodisc system connecting GFP or other components. This may already change the membrane conditions or structures of the nanodisc. Does the membrane structure change as the author expected?

Response: Using negative stained EM, we observed that our new tools did form the expected nanodisc structure (new Fig. 1D and E). Thus, the addition of GFP did not alter the membrane conditions of nanodiscs. To further demonstrate the performance of our new tools, we did control experiments using syt1 and sar1p mutants that can bind membrane, but are significantly less efficient than wild-type proteins in mediating remodeling (Hui et al., 2009 Cell; Huang et al., 2001 JCB). Indeed, these mutants elicited much lower responses from our sensor (new Fig. S5E). Thus, we believe that the nanodisc sensor did respond to the structural changes of lipid bilayers. In the revised manuscript, we have made the following changes:

“Further, C2A and C2A-C2B_{CLM}, mutants of syt1 that could bind membrane but were much less efficient in bending membrane³⁴, resulted in significant lower fluorescence responses (Fig. S5E).”

“In control experiments, the sar1p mutant (T39N) that was dominant negative in mediating vesicle budding⁴³, exhibited a ~6-fold decrease in the sensor response (Fig. S5E).”

3. What happens when the different concentrations of detergent are mixed in the buffer? A typical experiment of nanodisc replacement may still contain detergent in low concentration. Dose it will affect membrane formation or bind associated with the

membrane?

Response: In all of our experiments, we have purified our nanodiscs by gel filtration, and thus, the amount of remaining detergents should be negligible. However, we agree that detergent could affect the performance of the sensor. In the revised manuscript, we have performed titrations of the detergent DDM on the response of the sensor to syt1-mediated membrane remodeling (new Fig. S5F). The results showed that high concentrations of detergent disrupt the performance of our sensor, while low concentrations of detergent seem to have no effect. In the revised manuscript, we have made the following changes:

“We also found that the sensitivity of these nanodiscs could tolerate low amounts of DDM, perhaps because of the enhanced rigidity of the circularized scaffold protein (Fig. S5F).”

Reviewer #2 (Remarks to the Author):

Over the last decade, nanodiscs (NDs) have emerged as a powerful approach to reconstitute membrane proteins in a lipid environment. NDs are nanosized objects compatible with a large panel of applications. They have inspired many research groups to improve and optimize membrane scaffold proteins (MSPs) in order to make this membrane mimicking system even more versatile. In their study, Qian Ren and colleagues followed this path by designing circularized fluorescent NDs with the aim of developing new biosensors to monitor membrane binding and remodeling events. Although covalently-circularized NDs were already introduced a few years ago to stabilize large NDs, this study reports the complementation of the two halves of split GFP fused to the N- and C-termini of MSPs. Three MSPs of different lengths were used to generate circularized fluorescent NDs of different diameters. In order to link the GFP fluorescence emission signal to the membrane deformation mediated by protein binding to NDs, the authors optimized the GFP-MSP constructs. Then, as a proof of concept, they tested a panel of proteins that are capable of membrane remodeling. In addition to the development of these new NDs, the improvement of circularized NDs, which are stable and homogenous contributes to an improvement of this tool for the study of membrane proteins. Overall, the manuscript is clear and well written but many experimental details and clarifications in the figure legends deserved to be added.

Response: We are very grateful for the enthusiasm of this reviewer in our work. We hope that the reviewer finds the revised manuscript satisfactory.

My main comments are listed below:

- It is reported that all attempts to generate RFP-MSPs (via split RFPs or full-length RFPs) failed, unless SPY-spmcherry was used. The authors interpreted this result as a problem with RFP stability. However, it is unclear if the problem arises from a lack of RFP-MSP expression. It would have been interesting to indicate the levels of

expression of the different constructs. In the case of satisfactory expression levels of RFP-MSP, their ability to form NDs in the presence of lipids would be worth mentioning.

Response: We agree with the reviewer that more specifics on RFP-MSPs should be provided. We observed that most of these constructs did not produce measurable levels of proteins as shown by pull-down experiments and in-gel fluorescence imaging (new Fig. S2C). The ability of SPY- spmcherry1_{1-10/11}-MSP1D1 to form monodisperse NDs is less efficient than GFP-circularized MSP, likely because of its tendency to oligomerize. In the revised manuscript, we have made the following changes:

“To our surprise, none of the split mcherry in our trials complemented when fused to the N- and C-termini of MSP1D1 (Fig. S2C-E). Furthermore, fusion of several full-length RFPs with MSP was also not successful (Fig. S2C, F-I).”

“The efficiency of SPY- spmcherry1_{1-10/11}-MSP1D1 to form nanodisc is lower than SPY-spGFP_{1-10/11}-MSP1D1, as more than 35% larger particles were found by SEC analyses (Fig. S3A and B).”

- In Fig. S1L, only the SEC profile of SPY-spSFM-MSP (grey curve) has been commented to mention the low proportion of dimers (less than 10%). However, no comment is made on the profile of RFP-NDs (red curve) although it seems to show a higher proportion of dimers. Further comments on the results deserve to be given.

Response: Yes, we agree that RFP-NDs have a higher portion of large particles (~35%) and should be stated in the manuscript. In the revised manuscript, we have made the following changes:

“The efficiency of SPY- spmcherry1_{1-10/11}-MSP1D1 to form nanodisc is lower than SPY-spGFP_{1-10/11}-MSP1D1, as more than 35% larger particles were found by SEC analyses (Fig. S3A and B).”

- Based on the experiment performed in Fig. 2, the authors seem to suggest that the increase in FRET signal between PS containing GFP-ND and syt1-mcherry in the presence of calcium ions (Fig. 2B) results from a specificity of syt1 to PS since the FRET signal obtained in the presence of PC containing GFP-ND is much weaker. However, knowing that PS is negatively charged and PC is zwitterionic, calcium ions act as bridges and the experiment performed in Fig. 2C with PC cannot be interpreted as a control experiment to reveal syt1 specificity to PS. An increased in FRET signal would have been observed with any negatively charged surface. One note that the concentration of Ca²⁺ used is not reported in the Figure legend or in the Method section. More experimental details would be appreciated.

Response: We agree with the reviewer. Previous studies have shown that syt1 binds to many negative charge lipids in the presence of Ca²⁺. So, we changed our statement to the preferred binding of syt1 to negative charge lipids. In addition, Ca²⁺ concentration is 0.5 mM in all experiments. In the revised manuscript, we have made the following changes:

“In control experiments, we did not observe FRET using nanodiscs harboring only PC lipids (Fig. 2C), which is consistent with the preferred binding of syt1 to negative charge lipids^{31, 32}.”

“FRET was observed upon incubation of syt1-mcherry with isenND containing PS lipids in the presence (red) and absence (blue) of 0.5 mM Ca²⁺.”

- It is mentioned that different linkers between cpGFP and MSP have been screened in order to optimize NDs with the expected sensitivity to membrane protein binding. The given explanations appear rather speculative and although we don't really know why it ends up working, it seems to remain a black box. Here again, based on the level of detail provided in the Method section it appears impossible to reproduce the cloning work. More information on the sequences used for the iGlu-MSP construction is required to make this study of interest for other researchers.

Response: We agree with the reviewer that we are not sure why our design ends up working. In general, the linker region is critical for the development of many biosensors (Marvin et al., 2013 Nat Methods). Therefore, we screened a library of different linkers and identified one. We are doing molecular dynamics simulation studies to gain further insights into the sensing mechanism of our nanodiscs and hope to report our findings in the future. In addition, we are in the process of depositing our constructs on Addgene and have included the sequence of our constructs in the revised manuscript (new Supplementary Table 1). We have made the following changes:

“All the plasmids will be available at Addgene. The protein sequences of the engineered MSPs are described in Supplementary Table 1.”

Other minor comments:

Globally, the figure legends are not precise enough.

- *Figure 1B: mention in the legend if the gel is Coomassie-blue stained.*

Response: In the revised figure legends, we have clarified that the left panel is Coomassie-blue stained, and the right panel is in-gel fluorescence imaging. We have made the following changes in the figure legends:

“(B) SDS-PAGE (left, Coomassie-blue stained) and in-gel fluorescent imaging analysis (right) of circularized MSPs.”

- *Figure S2A: precise in the legend that the gel was illuminated at the GFP channel for in-gel fluorescent signal detection. Indicate if the experiment has been done in the presence of calcium. As syt1-ND shows smears on the gel, indicate with a box the area that has been used for signal quantification.*

Response: These gels were bought from BioRad and thus did not contain Ca²⁺ in the gel. However, the samples were prepared in the presence of Ca²⁺. We have also indicated with a box the areas that were used for quantification (new Fig. S4A). In the revised manuscript, we have made the following changes:

“(A) Representative native PAGE of syt1 binding to isenND containing PS or PC lipids by in-gel fluorescence imaging at GFP channel. ND (0.2 μM) were incubated with increasing concentrations of syt1 in the presence of Ca²⁺ (0.5 mM) at room temperature for 10 mins. Samples were then subjected to native electrophoresis using Mini-PROTEAN gels from BioRad. The encircled boxes were used for quantification by gel densitometry in panel B.”

- *Figure 3B: Precise in the legend the concentrations of GFP-NDs and that of membrane remodeling proteins. The addition of the full name of these latter and their*

molecular weight could be informative.

Response: In the original manuscript, we have provided some of these details in Methods. In the revised manuscript, we have now also included the concentration of GFP-NDs and membrane remodeling proteins with their name and molecular weights in the figure legends of the revised manuscript.

“Sensitivity of iGlu-MSP nanodiscs for different membrane remodeling proteins. iGlu-MSP nanodiscs (0.1 μ M) were incubated with the indicated protein (1 μ M) at room temperature for 10 min, followed by fluorescence spectroscopy analyses. EIIA^{Glc} and BSA were used as controls. The cytosolic domain of synaptotagmin-1 (syt1), 37 kDa; sar1p, 21 kDa; complexin-2 (cpx), 15 kDa; α -synuclein (syn), 14 kDa; α -hemolysin (HA), 36 kDa; HIV-1 fusion peptide (HIV_FP), 2.9 kDa; SARS2 fusion peptide (SARS_FP), 4.6 kDa; Magainin2 (Mag2), 2.5 kDa; EIIA^{Glc}, 22 kDa; BSA, 67 kDa.”

- Figure 3C: for the blue and black curves present on the fluorescent spectra, specify the conditions to which they are associated.

Response: We have specified these conditions in the figure legend of the revised manuscript.

“Representative fluorescence spectrum of iGlu-MSP nanodiscs in the absence (black) and presence (blue) of syt1 at the indicated conditions. Data are shown as mean \pm s.d., $n \geq 3$ independent experiments.”

Reviewer #3 (Remarks to the Author):

Ren et al., describe a fluorescence-based assay for detecting membrane protein binding events on circularised nanodiscs. This methodology which relies on split-GFP technology could also be used to circularise (and detect formed) nanodiscs could constitute a standard tool in membrane protein biophysics and structural biology. The method could be used for sample condition, or protein reconstitution optimisation and to monitor biological (related to membrane binding) processes. The authors test their isenND method on various pro- and eukaryotic proteins, and demonstrate its ease of use and applicability. They also highlight the potential of their method as being fast and reliable, relying on readily accessible and affordable equipment.

The manuscript is well written and I see no major flaws with neither the reported data reproducibility or its analysis. Circularised nanodiscs are becoming increasingly important in membrane protein biochemistry, biophysics and structural biology. The pursue for highly stable nanodiscs to accommodate large protein complexes for analysis by CryoEM, EPR (PELDOR or DEER) or smFRET, makes this method an important tool for modern structural biology, where there is now an increasing need for studying membrane proteins in their native lipid environment and thus acquiring data within stable, monodisperse, circularised nanodiscs of any size. In particular, for larger nanodiscs described here (NW30/50), which are more challenging to form, stabilise and control, due to their highly dynamic nature, a tool such as this, could allow for the detection of specific protein binding events and interactions with membranes, using

readily accessible equipment.

Response: We are gratified that the referee found our study has the potential to advance the field. We hope that the reviewer finds the revised manuscript satisfactory.

Points to address:

1) On figure 1C (bottom). If NW50 elute at “void” volume of a Superose 6 column, how could the NW50 monodispersity be judged? Perhaps another column with lower resolution (for larger particles though) may be able to discriminate among NW50 sizes/states, e.g. Sephacryl S-500? The authors present supporting DLS data (1D) for NW50 size distribution, however it would be good to support DLS conclusions with higher resolution SEC, given that “large” NW50 nanodisc formation is more of a “bigger” issue (significantly broader distribution in DLS), compared to forming smaller nanodiscs. Is it possible that for NW50 two or more species co-exist? If yes, this could have implications for high resolution structural studies, such as cryoEM.

Response: We agree with the reviewer that NW50 nanodiscs are approaching the limitation of the Superose 6 column. It is thus difficult to estimate the quality of these large nanodiscs by gel filtration. In the revised manuscript, we have performed negative stain EM and found that these large nanodiscs were relative monodisperse (new Fig. 1D and E). However, the size distribution of NW50 NDs is a lot broader than small ones. So, structural studies using these large NDs might be challenging. In the future, we will start to use the Sephacryl S-500 column to purify NW50 NDs. Unfortunately, we currently do not have this column as a young lab and could not find one on our campus. Everything is still back-ordered because of the current pandemic. We will definitely use them in the future characterization of large nanodiscs and further assess their potential utility in single-particle cryo-EM studies. Nevertheless, we believe that the expanded lipid surface area of NW50 NDs will be useful for the biochemical characterization of protein-lipid interaction using fluorescence-based approaches. In the revised manuscript, we have made the following changes:

“Using SEC and EM analysis, we indeed found that spGFP_{1-10/11}-NW30 and spGFP_{1-10/11}-NW50 could form 29 and 48 nm large circularized nanodiscs (Fig. 1C-E), respectively. These values were also corroborated by DLS measurements (Fig. S1).”

2) On figure 2. Could the authors clarify what the “independent” experiments mean? Independent nanodisc batches for each, protein preparations, etc.

Response: Throughout the work, these are three independent nanodisc batches and protein preparations. In the revised manuscript, we have defined this issue in the statement of quantification and statistical analysis:

“Independent experiments were defined as replicates performed with separately prepared batches of protein and nanodiscs.”

3) Again in Figure 3 as in figure 2 (previous comment). Please specify what “independent” experiments account for

Response: As mentioned above, these are three independent nanodisc batches and protein preparations.

REVIEWERS' COMMENTS:

Reviewer #1 (Remarks to the Author):

Authors have made all required revision. I am happy with the revision.

Reviewer #2 (Remarks to the Author):

The authors have submitted a revised version with additional data, which definitely strengthens their study. However, I suggest that they be more specific in describing some of the results.

- Fig. S2C: RFP-fused MSP constructs expressed in *E. coli* were analyzed by SDS-PAGE. According to the legend, the samples were prepared by pull-down experiments but the experimental procedure is not described. So, it is difficult to know what was done and to follow the result. Also specify the MW of each construct to help the reader know where to look at on the gel.

- Fig. S5E: in contrast to the other *syt1* mutants, the effect of C2ACLM-C2B is not commented in the main text. For this mutant, a significant defect in membrane tubulation is reported in ref 34, whereas in the context of nanodiscs the strong fluorescent signal seems to indicate no effect of this mutation. An attempt to explain this discrepancy deserves to be given.

Reviewer #3 (Remarks to the Author):

The changes made and explanations provided are satisfactory. I am supportive of the manuscript being accepted without any further revision(s).

Reviewer #1 (Remarks to the Author):

Authors have made all required revision. I am happy with the revision.

Response: We are very grateful for the enthusiasm of this reviewer in our work.

Reviewer #2 (Remarks to the Author):

The authors have submitted a revised version with additional data, which definitely strengthens their study. However, I suggest that they be more specific in describing some of the results.

- Fig. S2C: RFP-fused MSP constructs expressed in E. coli were analyzed by SDS-PAGE. According to the legend, the samples were prepared by pull-down experiments but the experimental procedure is not described. So, it is difficult to know what was done and to follow the result. Also specify the MW of each construct to help the reader know where to look at on the gel.

Response: We apologize for this oversight and have included a description of the pull-down experiment in the Methods of the re-revised manuscript. In addition, the MW of each construct is provided in the figure legends. We made the following changes:

“SDS-PAGE of the designed RFP-MSPs pulled down from bacterial cells stained with Coomassie blue (left) and in-gel fluorescence imaging at the mcherry channel (right). **1**, spmcherry1_{1-10/11}-MSP1D1(51 kDa); **2**, spmcherry3_{1-10/11}-MSP1D1(51 kDa); **3**, mcherry-MSP1D1 (51 kDa); **4**, GECO-MSP1D1 (52 kDa); **5**, LSSmcherry-MSP1D1(53 kDa); **6**, RDSmcherry-MSP1D1(53 kDa); **7**, SPY-spmcherry3_{1-10/11}-MSP1D1(65 kDa); **8**, SPY-spmcherry1_{1-10/11}-MSP1D1 (65 kDa).”

“To screen the expression level of MSPs, cells were grown in 20ml culture to OD600 ~0.5 and induced with 0.1 mM IPTG at 16 °C overnight. Bacteria were harvested by centrifugation at 6000 rpm for 10 mins, resuspended in Buffer A and lysed by freeze-thaw plus the addition of 1% Triton at 4 °C overnight. Cell lysates were clarified by centrifugation at 10, 000 rpm for 20 mins. The supernatants were incubated with 100 μL His60 Ni Superflow Resins for 20 min with gentle shaking at room temperature. Samples were loaded onto empty micro Bio-Spin chromatography columns (BioRad #7326204), followed by two times wash using buffer B. Proteins were eluted in buffer C and subjected to analysis by SDS-PAGE. ”

- Fig. S5E: in contrast to the other syt1 mutants, the effect of C2ACLM-C2B is not commented in the main text. For this mutant, a significant defect in membrane tubulation is reported in ref 34, whereas in the context of nanodiscs the strong fluorescent signal seems to indicate no effect of this mutation. An attempt to explain this discrepancy deserves to be given.

Response: We are not sure about the discrepancy of C2A_{CLM}-C2B mediated membrane remodeling between our data and the data in ref. 34. The most likely reason is the methods used in the two studies. Our sensor responds well to minor membrane remodeling reactions, whereas negative stain EM in ref.34 is more sensitive to drastic changes of lipid bilayers. In addition, the quantification of membrane bending in ref. 34 is only for tubulation. However, membrane remodeling events can also cause generation of small vesicles, which is not quantified in ref. 34. It seemed that much more small vesicles were observed in the presence of C2A_{CLM}-C2B than C2A-C2B_{CLM} (Fig. 2, ref. 34), indicating that this mutant might have higher membrane remodeling activities. Nevertheless, our explanations are all speculative, and we believe that structural elucidation of membrane remodeling by syt1 is required to reconcile these conflicting observations in the future. In the re-revised manuscript, we have attempted to discuss this issue and made the following changes:

“In contrast, C2B, which retained the ability to bend membrane, caused a high fluorescence response as the wild-type protein. Interestingly, previous studies showed that C2A_{CLM}-C2B is less active in membrane bending than the wild-type protein using negative stained EM³⁴, while exhibiting similar responses in our assay. We suspect that this discrepancy is most likely due to the sensitivity of the two assays, although the exact reason is unclear and will require structural elucidation of membrane bending by syt1.”

Reviewer #3 (Remarks to the Author):

The changes made and explanations provided are satisfactory. I am supportive of the manuscript being accepted without any further revision(s).

Response: We are very grateful for the enthusiasm of this reviewer in our work.